# Clavis: An open and versatile identification key format

Wouter Koch[1,2]*, Hallvard Elven[3◉], Anders G. Finstad[1◉]

**1** Department of Natural History, Norwegian University of Science and Technology, Trondheim, Norway, **2** Norwegian Biodiversity Information Centre, Trondheim, Norway, **3** Natural History Museum, University of Oslo, Oslo, Norway

◉ These authors contributed equally to this work.
* wouter.koch@artsdatabanken.no

## Abstract

The skills and knowledge needed to recognize and classify taxa are becoming increasingly scarce in the scientific community. At the same time, it is clear that these skills are strongly needed in biodiversity monitoring for management and conservation, especially when carried out by citizen scientists. Formalizing the required knowledge in the form of digital identification keys is one way of making such knowledge more available for professional and amateur observers of biodiversity. In this paper we describe Clavis, an open and versatile data format for capturing the knowledge required for taxon identification through digital keys, allowing for a level of detail beyond that of any current key format. We present the format independently from any particular implementation, as our aim is for Clavis to serve as a basis for interoperable tools and interfaces serving different needs and actors.

**Data Availability Statement:** All relevant data are within the manuscript and its Supporting information files. Additionally, all files related to the manuscript have been deposited on Zenodo (DOI: 10.5281/zenodo.7209011).

## Introduction

Distinguishing biological taxa from one another is a necessity in biodiversity monitoring. As species are going extinct at an unprecedented rate [1, 2], we need to monitor as much of nature as we can to identify population statuses and trends. Meanwhile, research and management is currently facing a "taxonomic impediment", where taxonomic knowledge is gradually disappearing from the scientific community [3].

Paradoxically, while taxonomic expertise is becoming increasingly scarce, species observations are being reported like never before. The bulk of the new observational data originates from citizen science, in which observations are made by non-professional volunteers [4]. There are large taxonomic biases in the data collected from these sources. For example, of publicly available data from the Global Biodiversity Information Facility (the largest aggregator of biodiversity observations in the world [5]), some 67% of over 2 billion observations pertain to birds [6]. This means that less charismatic species, and species that are more difficult to identify, generally remain severely under-reported [7]. Many such species play important ecological roles or can serve as indicators of broader ecosystem health. The fact that few observers are able to reliably identify these species leads to their underreporting, obscuring important trends from the view of science and management [8].

**Funding:** WK is funded by the Research Council of Norway (https://www.forskningsradet.no), grant no. 272947. The funders had no role in study design, data collection and analysis, decision to publish, or preparation of the manuscript.

**Competing interests:** The authors have declared that no competing interests exist.

The knowledge needed to identify species resides in large part in the heads of taxonomic experts. This invaluable experience is slowly disappearing as newly educated biologists are, to an increasing extent, trained with focus on skills in genetics, data management, and biodiversity informatics rather than traditional taxonomy. This training, and an academic reality of shorter periods with funding and more temporary contracts, gives fewer researchers the opportunity to invest the years of study needed to become truly knowledgeable with regards to any substantial taxon [9].

Identification keys are one of the most important tools used by taxonomists to identify taxa, as well as to share with others the information needed for distinguishing taxa. No taxonomist studying a large taxon in depth can get around identification keys, and keys are often the starting point when trying to learn a new organism group. Identification keys can, however, be quite challenging to use, especially for novel users, and their use often represents a significant barrier for the prospective learner. For this and other reasons, the use of identification keys is receiving less focus in most current biology curricula. Furthermore, as taxonomic knowledge is becoming increasingly fragmented, existing identification keys in literature are gradually becoming outdated as taxonomies are changing according to new insights.

These issues are exacerbated in the context of citizen science. Whereas professional taxonomists have an incentive to learn to use a key, the more casual citizen scientist can easily be put off by the steep learning curve. Citizen science, with its unrivaled amounts of data generated, but also with added concerns about the reliability of the taxon identifications [10, 11] and the pronounced over-representation of more charismatic taxa [12–14], stands especially much to gain from good identification keys with a low user threshold.

Digital identification keys address these challenges in a number of ways. The formalized and machine-readable way in which the data are stored allows for easier revision, as well as multiple options for their display to the end user, tailored to their level of experience. Digital keys allow for the representation of more complex relationships between species' characteristics and identifications inherent to biological complexity, than cannot be easily represented in the linear form of most traditional, paper-based keys. With a suitable interface, digital keys open up the cumulative experience from taxonomic experts to a broader public like citizen scientists, while potentially being more reliable and educational than other identification methods such as automated image recognition. In contrast to paper-based keys, digital keys can also be combined with one another, integrating the information from several separate keys into one seamless user experience. And conversely, a digital key can easily be limited to just a subset of the taxa it covers, for instance only taxa belonging to a certain habitat or geographic range. Another benefit of digital keys is the possibility of saving the user input and the key together with the observational data, as a means of transparent identification that can be reviewed at any time in the future for quality control or re-evaluation in the case of new taxonomical insights.

Having a fully open and well-defined, platform-independent format to store key knowledge is essential in ensuring that identification keys and the tools needed to display and create them remain both interoperable, interpretable across platforms, and freely available to all. A number of digital identification key formats exist [15–17], but these come with a number of limitations in what they can represent, their ease of use, openness, etc. A well-defined format addressing this alleviates the technical burden in capturing taxonomic knowledge for future use (see Table 1).

In this manuscript we document Clavis; an open format for identification keys, aiming to cover the aforementioned requirements in an open and lightweight manner, serving as a way to store and exchange crucial taxonomic knowledge. Clavis is intended to capture all the requirements of traditional keys while adding the flexibility and complexity possible with

**Table 1. Comparison of the DELTA and Clavis format.**

|  | DELTA | Clavis |
|---|---|---|
| **Main purpose** | Generation of textual descriptions for classification | Storage of identification key knowledge |
| **Support in modern programming languages** | Not supported (custom code) | Natively supported |
| **Hierarchical taxonomy** | Not supported | Supported |
| **References to taxonomic services** | Not supported | Supported |
| **Multilingualism** | Not supported | Supported |
| **Multimedia elements (internal or external)** | Not supported | Supported |
| **External descriptions** | Not supported | Supported |
| **Georeferencing** | As non-machine readable text only | As machine readable Geo-JSON |
| **Numerical character values** | Only int or float | Any step size and unit |
| **Logic for dependent characters** | Limited | Support of complicated logic |
| **Sub-endpoint** | Not supported | Supported |
| **Frequency** | As non-machine readable text only (e.g. "<Rare>") | As machine readable values |
| **Metadata (organizations, persons, roles, license, references)** | Not supported | Supported |

Note the difference in purpose; Clavis does not aim to replace DELTA in all its use cases. We do not include a comparison with Lucid, as it is a closed format that can thus not be evaluated.

digital keys. This means that the format is well suited for representing most, if not all, existing keys, both paper-based and digital, as well as for designing new keys. The name Clavis means "key" in Latin, and is a recursive acronym for <u>C</u>lavis <u>L</u>ightweight <u>A</u>nd <u>V</u>ersatile <u>I</u>dentification <u>S</u>chema.

This article describes the Clavis format itself, independent from any specific code implementing the format in an end-user interface. The choice to focus solely on the format here is a conscious one. Analogous to describing a new statistical method versus its use in a particular R package, conflating for example a key viewer with the data format in which the key data are stored can hamper the scrutiny of either. Bugs and omissions in the interface are not related to the data format, and the specifics of the interface may be such that aspects of the format that are suboptimal remain hidden from view. There can be many different tools supporting Clavis, each serving a particular need or user group, such as editors for making keys and different end user interfaces for rendering them. Focusing on any such tool here would undermine the general nature of Clavis as a data format, and obfuscate strengths and shortcomings of the format versus a tool that uses the format, hindering its adoption and improvement. Additionally, software packages have limited lifespans, and are generally outlived by the formats they build upon. Nevertheless, the central aspects of implementing Clavis are discussed in this manuscript in a generic way, and code handling the business logic of keys described using Clavis will be published separately at a later time.

Descriptions of each of Clavis' data types and emerging properties are described in this manuscript. Notation examples are provided in the supporting information (S2 File), in which all these concepts are illustrated using fictional taxa that are not subject to taxonomic debate or change. This ensures that our examples will not lead to confusion as they will not become at odds with future taxonomic insights, as any biological example eventually would, while exhibiting the required complexity for a demonstration of the more intricate features of the format.

## Materials and methods

Identifying species can be notoriously difficult. The main challenge with designing good identification keys is that nature is often not as clear cut as one would like, at least for classification

purposes. Ideally, taxa would be readily separable by discrete traits that showed no variation within each taxon, but clear differences between taxa. In reality, keys need to account for significant variation within the taxa of interest, as well as often significant overlap in traits between the taxa. Even at the species level, a key must deal with within-species differences in the form of different sexes, morphs, castes, ages/stages, and/or geographic or seasonal variants, as well as the general phenotypic variation between individuals and the possibility of hybridization between species. Moreover, the relevance of certain characters can depend on the states of other characters. For keys to higher taxonomic groupings, such as genera or families, these challenges from intra-group variation are only exacerbated. Furthermore, there is often a limited number of characters to choose from for distinguishing taxa, and available characters are often continuous rather than discrete, such as traits pertaining to size or proportions. In the context of keys, continuous characters are usually dealt with by forcing them into artificial, discrete categories.

In addition to the challenges with the characters themselves, users face the challenge of observing the necessary traits and interpreting them correctly. Given the right tools, however, much can be done by the key producer to improve the user experience and reduce the risk of incorrect identifications. Illustrated keys are far easier to use than text-only keys, and the inclusion of good illustrations greatly reduces the risk of misinterpreting characters. Other useful aids are explanations of scientific terms and hints for how to best observe the relevant traits. Providing fewer alternatives to choose from is mentally less demanding of the user, and keys that combine several traits in the same alternative (e.g. "legs black AND head hairy") are easier to use than keys that describe just one trait at a time, because the user gains more confidence in the answer by being able to check several traits together. Many keys also include more helpful information of a more secondary nature with the alternatives ("Legs black. Mostly in the mountains").

A data format for sharing and rendering taxonomic information in the form of a key must be able to handle all these aspects of keys and more. It must handle all the discrete and continuous variation within and between taxa inherent in biological systems, and for the organization of both taxa and characters both linearly and hierarchically. It must allow for the inclusion of media elements, links to supplementary information, different language options and other aids in the key, and it must provide the key producer with the necessary control tools to ensure that the user faces a key that behaves correctly.

Identification keys exist in many forms, but all are built on the same basic principle. The user is asked a number of questions about the entity being identified, and by answering these questions, taxa are excluded from consideration until (ideally) only one taxon remains, which is the result of the identification process.

Traditional single access keys consist of a decision tree with a single fixed path from beginning to end for each possible outcome. On choosing an answer from the alternatives for each of the questions, the user is directed to the next question to be answered, until they end up with a result taxon. Such keys can be either dichotomous, meaning that each question always has only two alternatives, leading to a bifurcating decision tree, or they can be multichotomous, meaning that each question may have more than two alternatives. One upside of single access keys is that they are easy to represent on paper. A drawback is that a user has to go through all questions in a specific order, having one and only one question to answer at any given time. This means that one cannot use only part of the key for an arbitrary subset of taxa, nor easily go back and redo a question. But most importantly, if a question cannot be answered, there is no way to proceed and no way of knowing which taxa remain.

An alternative approach is the multiple access key. Such keys are typically stored as matrices where taxa and questions (characters) are stored as rows and columns in a tabular format, with

cell values linking the taxa to their characters. Each character may have two or more possible answers (states). In a fully populated matrix key, every character is assigned a value (i.e. is scored) for every taxon. The user then has access to all the questions/characters at once and is not required to answer them in any particular order. Choosing any alternative for any of the questions will exclude from consideration all those taxa that are not compatible with that alternative, bringing the user a step closer to the answer. The upside of this approach is that the user has several paths to the answer and can choose to avoid questions that are difficult or impossible to answer. The downside is that the number of choices can easily be overwhelming, and it is largely left to the user to try to choose the best path. Also, in real life situations it is very rare to find a set of characters that are both possible to score in a meaningful way for all the taxa in the key, and sufficient for distinguishing between them. More often, only some characters will be possible to score for all taxa, whereas others may be essential for distinguishing certain taxa in the key, yet be completely inapplicable to others.

To address this, a matrix can be filled out sparsely instead, meaning that not all characters are scored for all taxa. The interface can then display only those characters that are in fact scored for all the taxa currently under consideration. As taxa are being excluded by the user's choices, further characters that are scored for all the remaining taxa will become available for answering. With this approach, it is possible to make a matrix key that behaves equivalently to a single access key. But the key can also be made to contain several distinguishing characters for any set of taxa, so that the user has more than one choice at each step and is thus not restricted to having to answer only one question at a time in a specific order.

The sparse matrix approach is more powerful than both the single access and the full matrix approach, in that both latter approaches are subsets of what the sparse matrix can represent. An important advantage of either matrix approach over the single access key is that the key can easily be restricted to only a subset of the taxa. This means that the user will not have to traverse the whole key if they can exclude one or more taxa a priori. This sparse matrix approach has been the approach of the precursors of Clavis in our work for several years. We increasingly found, however, that there is much additional information regarding taxa, characters, and the relationships between the two, which cannot be easily represented in a matrix format. For instance, a matrix is not very well suited for dealing with polymorphism, or with non-discrete characters such as measurements. Also, by its very nature a matrix links a list of characters to a list of taxa, but has no easy way to add structured data to specific parts of the content, such as a hierarchical taxonomy, geographic information pertaining to specific characters, taxa or relations between the two, etc. Thus, representing taxonomic knowledge in a tabular format restricts the complexity of the information that can be stored, and greatly complicates that code that interprets such matrices.

The method of storing and exchanging taxonomic knowledge described here, is developed as a non-tabular multiple access key. Building on years of experience with capturing keys for a range of taxa in sparse matrices, one of the core concepts of defining Clavis has been the representation of uncertainty. By allowing non-binary traits (i.e. taxon X *can* have property Y, but not necessarily), interdependencies (character X is relevant *only* if the answer on character Y is Z), numerical ranges (species X is between Y and Z mm long) as well as the possibility to make any number of custom groupings and subunits in the taxonomy, Clavis can represent multiple layers of nuance and uncertainty. For a key that is guaranteed to give one and only one taxon as an answer, there will logically need to be at least one set of unique features for each taxon, but such completeness is not a requirement in Clavis. One can make a valid Clavis key that can in some cases result in a list of remaining taxa that, given the properties of the specimen at hand, can not be reduced further. Such keys can still be valuable assets for the end user.

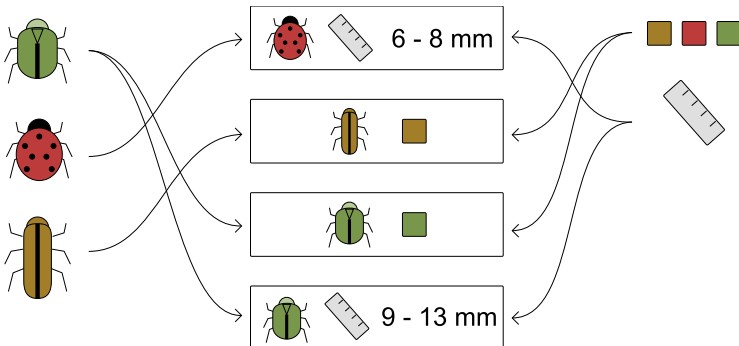

**Fig 1. The core concepts of any Clavis key.** Taxa (left) and characters (right) are connected through statements (center). Each statement refers to one taxon and one character, linking the two by a value, usually one of the possible states of the character, or a numerical value or range.

The core of any Clavis key is a collection of statements. A statement links a taxon to a character, and specifies a state or numerical value the taxon has for that character (see Fig 1). E.g. the color (character) of species x (taxon) is red (state), or the length in millimeters (character) of genus x (taxon) is 6–8 (numerical range).

Additional information can be linked to taxa, characters, states, and statements, capturing knowledge and relationships that cannot be represented in a tabular form in a practical way. Such information includes media elements, textual descriptions, and contextual information such as geographic scope. As the Clavis format is designed to support all the possibilities of the linear key and matrix key, any existing key using these approaches can be transcribed to it.

Clavis compliant keys are written in JSON (JavaScript Object Notation) [18]. JSON (pronounced "Jason") is a widely used open format supported natively by all modern programming languages. Clavis itself is a JSON-schema [19], a formal definition of the structure and content of a valid Clavis key file. JSON-schemas are machine readable and can be used to automatically validate the compliance of a JSON file in modern code editors, highlighting any issues that need to be solved. While it is possible to manually write the JSON of a Clavis-compliant key (as was done for this manuscript), it is generally not practical. In order to facilitate key creation and editing, one would generally provide taxonomists with a key editing interface that handles the generation of the JSON file.

## Implementation

As an exchange and storage format, Clavis does not dictate how it is to be implemented. Different interfaces can build upon it differently, depending on the purpose of the interface and the intended user group. To this end, interfaces serving to edit or display keys may omit certain non-essential functionality supported by Clavis as a whole. One could for example create a key editor not supporting media files or multilingualism. Other aspects of the format are crucial, however, and require support and unambiguous interpretation. These aspects are discussed in the current manuscript.

Identification is a matter of excluding taxa, and is done by letting the user select a state or numerical value for characters. Each time the user provides a new fact in this way, all taxa with conflicting statements are excluded (see Fig 2). The user can also be given the option to exclude a state rather than affirm one if there are more than two states to choose from.

Whenever taxa are excluded, characters that were previously hidden may now have become relevant if all the remaining taxa have a statement pertaining to it, as illustrated in Fig 2. These

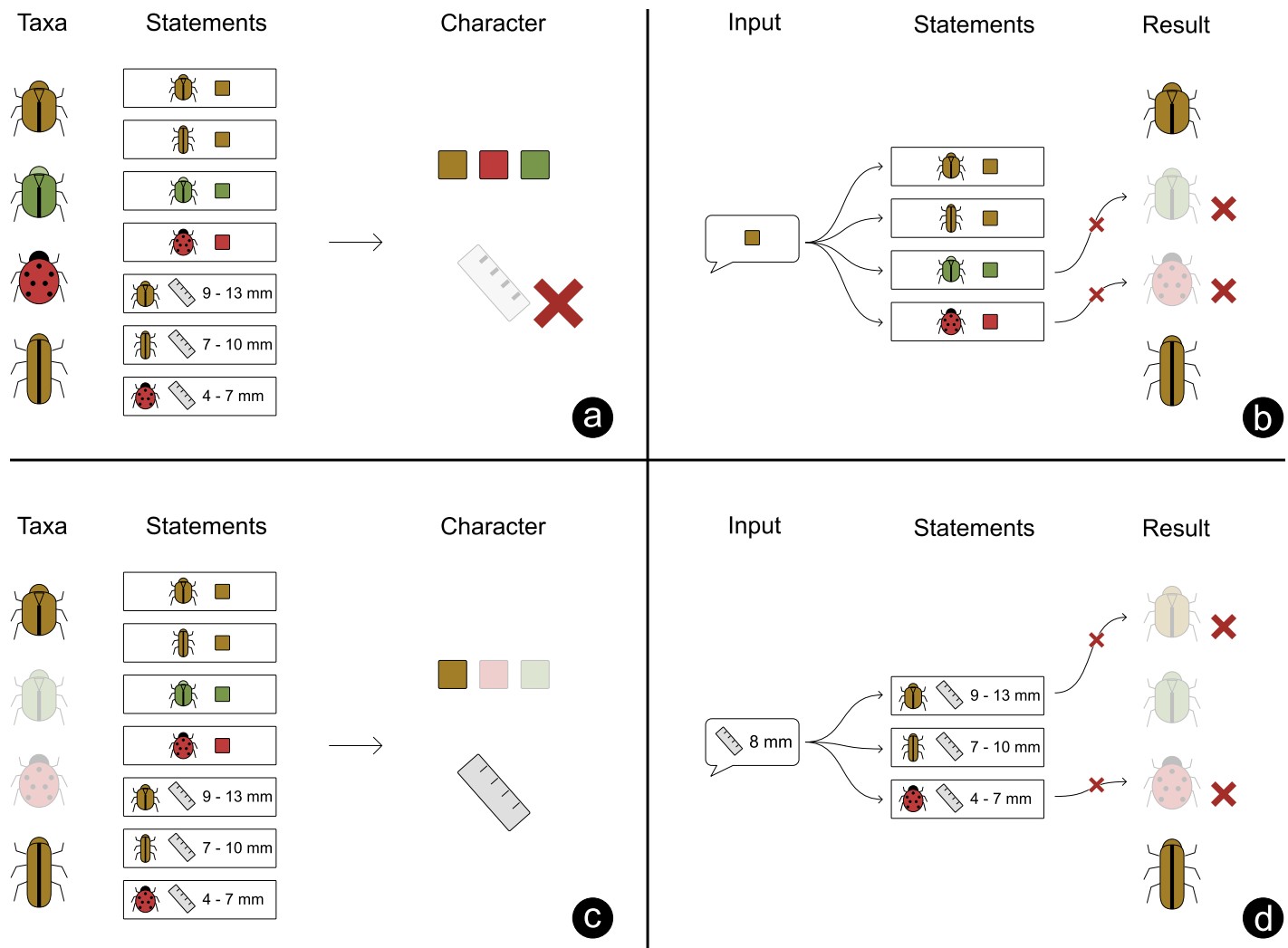

**Fig 2. The identification of a specimen.** This example contains four taxa and two characters (color and size). The taxa and characters are connected by statements, specifying a character state connecting the two. a) Initially all four taxa are possible outcomes. At this point, only the color character is made visible to the user, as only this character has a statement for all four taxa. The size character is hidden as it lacks a statement for one of the four taxa. b) the user gives an input about color. Based on this, two taxa with conflicting statements are excluded. c) the size character becomes relevant as it has statements connecting it to both the taxa now remaining. This character is made visible to the user. d) the user gives an input about size. Based on the related statements, one additional taxon is eliminated, leaving only one taxon. This is the end result.

characters should then be made visible to the user. The act of excluding taxa will also affect which of the possible states of a given character are relevant, even if it has not been answered directly by the user. If all the remaining taxa are known not to have a certain state, that state can be disabled. In such cases, the state still needs to be visible to the user as it provides context to the remaining alternatives, but it should not be possible for the user to select it. Characters that have not been answered, but that only have a single possible answer for all remaining taxa, can be hidden. In all cases, only characters that are linked by statements to <u>all</u> the currently non-excluded taxa should be shown to the user. Clavis also allows for the inclusion of controlling statements called logical dependencies, which can be used to only enable characters based on answers given on one or more other characters.

Things become more complicated when users are allowed to undo previous answers. Undoing an answer might render a character that was subsequently answered irrelevant again. To prevent answers to this now irrelevant character from eliminating taxa as possibilities, an implementation needs to be able to remove or ignore the user input on characters that have been rendered irrelevant.

The provided schema ensures that a key adheres to the Clavis format in a technical sense, but it does not ensure that any compliant key is logically complete and consistent. One can make a valid key that contradicts itself or that does not contain sufficient information to reliably distinguish between certain taxa. It is advisable to implement checks for this in any key-editor, and when keys by third parties are to be displayed, to also ensure that such cases are handled by the end-user interface.

To illustrate the features supported by the Clavis format, we here describe every feature and give some example uses. Examples in Clavis-compliant JSON are provided in the supporting information (S2 File), where examples relate to fictional creatures that are not subject to taxonomic debate or change. For this we have chosen Pokémon as they appear in the mobile game Pokémon Go, as these are clearly defined, yet exhibit sufficient complexity to demonstrate the various features of the format [20]. This also allows us to define and introduce all layers of uncertainty and non-binarity needed to illustrate those features, without risking future taxonomic insights rendering such considerations obsolete. Keys for natural taxa would rarely use as much of the possible functionality as we demonstrate here in a single key. The key is valid in accordance with the Clavis JSON-schema, but it does not contain all the information needed to distinguish between all of the Pokémon mentioned in it. For this, we also provide an example of a non-artificial key that does identify all taxa in it, but that does not aim to demonstrate all possible features that Clavis supports. This key covers all the species of titmice in Norway and can be found in the supporting information (S3 File).

## Results and discussion

The main components of the Clavis JSON-schema are the taxa that the key is designed to distinguish between, the characters describing their properties, and statements connecting taxa to characters through states or numerical values. Additionally, the schema defines a number of metadata fields, as well as custom data types, that can be referred to in various places in the key, such as when referring to a person as a creator of a picture, or linking a picture to a taxon, a character or a state.

### Format overview

An overview of the elements in an identification key defined by Clavis. Elements with an asterisk (*) are mandatory.

**Key metadata**

**Title**\* The name of the key

**Schema**\* The url of the version of Clavis that the key adheres to

**Media** A media element for the key, such as a logo or icon

**Description** A short, extended, and/or external description of the key

**Audience** A description of the intended audience

**Source** Name and/or link to the source the key is based upon

**Geography** Polygon and/or name of the region where the key is valid

**Roles** Primary contact, creators\*, contributors, publishers of the key, as references to persons and/or organizations

**License**\* An url to the license text the key is licensed under

**Language**\* The language(s) the key supports

**Dates**\* The dates the key was created and last modified (the version)

**Identifier**\* An id for the key, that remains stable over versions

**Url** Where the key is hosted, so that new versions can be retrieved

### Key content

**Taxa**\* A flat or hierarchical list of taxa the key is able to distinguish between. The goal of the interface is to eliminate all but one taxon.

**Characters**\* A list of characters used to distinguish between taxa. These are the questions that are presented to the user. A character can be categorical or numerical. When categorical, it has a list of states that are the relevant alternatives for this character.

**Statements**\* Elements connecting a taxon to a character through a state or numeric value. This is the core knowledge captured by the key: which taxa have which states or numerical values for which characters, thus distinguishing them from one another.

### Data types

**Person** The name\*, contacts, media elements, and affiliations of a person.

**Organization** The name\*, contacts, and media elements of an organization.

**Taxon** The scientific name, author, vernacular name, label, media elements, rank of a taxon. Info on whether it serves as an end-point or not. It can have a set of children, which are the underlying taxa. An external reference can define where info is to be retrieved from. It may have a geographic distribution as an object or external service, to assess where it occurs. A follow-up key as a url or external service reference can provide info on where a more detailed identification can be done.

**Character** The name\* and states describing a property a taxon can have. It can have media elements and descriptions clarifying the character, and a user requirement. A logical premise can specify what other user input has to be given before the character may be presented. A character can be of the types "exclusive" (default, multiple choice where options exclude one another), "non-exclusive" (multiple choice with multiple answers possible), and "numerical" (the answer is a number). A character has states defining the possible answers if it is not numerical, and a min, max, unit and step size if it is numerical.

**State** A possible answer for a non-numeric character. Has a title and/or one or more media elements, and may have descriptions further clarifying the state.

**Statement** A connection between a taxon and a character through a value (either a state or a numerical range). If it refers to a numerical state, the value can be a single value or a range. A statement also defines how frequently and in which context the taxon has this value for that character. A statement can have any frequency from 0 to 1, to indicate that the taxon always (1), never (0) or in some cases (values between 0 and 1) has this value for the character. Statements can contain references to a geographic distribution (or a service providing

one), defining where this statement is valid. Media elements and descriptions can be added describing the relationship between the taxon and character in more detail.

**User requirement** A user requirement can have a title, media elements and descriptions describing certain skills, equipment or other requirements needed to evaluate a character. It can also have a warning text to alert the user of these requirements. It can be used to guide a user where necessary, or help the user decide whether to skip more challenging characters.

**External service** A reference to an external service and its documentation. The creator of an interface can then choose to implement this external service so that e.g. taxon names are retrieved from an up to date repository by the provided stable identifier.

**Media element** A media element contains the data needed to display multimedia files. It can refer to different versions for different languages, and can have different versions for different media dimensions. It contains the required metadata such as width and height (images and video), length (sound and video), as well as creators, contributors, publishers and a license. Files can be urls to where the correct version of the file is to be found, or directly contain a base64 or svg encoded file. It supports external services to retrieve data from elsewhere.

While all core concepts are described with examples, not every combination of concepts are exemplified here. So while both multilingualism and descriptions are demonstrated, there is no example of multilingual descriptions. Such features do follow the same logic as the examples provided, and are all specified in the JSON schema.

## Key metadata

Very few parameters are required on the top level of the key. Apart from the content of the key (taxa, characters, and statements linking the two), a key is expected to refer to the version of the schema with which it complies, a title, language, license, a creator, the date at which it was last modified and an identifier that is to be kept stable across versions. As the creator has to be a reference to a person entity, at least one person needs to be defined as well.

*Example*: *See lines 2–10 in the supporting information (S2 File) for the corresponding JSON.*

## Taxa

Eliminating all but one taxon is the goal of the key, and taxa are the units that all characteristics are connected to. Taxa can be provided as a flat list, but they can also be structured hierarchically, for instance adhering to their phylogeny. Statements can in such a hierarchy be connected to higher levels in the hierarchy (higher taxa), reducing a lot of the repetition of traits shared within a taxon that one would get when using a flat list of taxa. If a statement is tied to a higher taxon in the hierarchy, it is implied that all the underlying taxa share the same statement.

In addition to biological taxonomic units, one can also define sub-groups within a taxon. Examples of such sub-groups can be different sexes, morphs, or species complexes within a species. Like the main taxa, the sub-groups can also be arranged hierarchically. Contrary to regular taxa, such subdivisions of taxa are not standalone taxonomic units. As such, they do not have their own scientific name but rather a label that adds specificity to their parent taxon. For example, the label of a sub-group specifying the sex of species X would simply be "♀" rather than "Species X ♀" as it already relates to the parent taxon "Species X". For a default sub-group of a taxon, the label can be an empty string. For instance, if a normally winged species has a

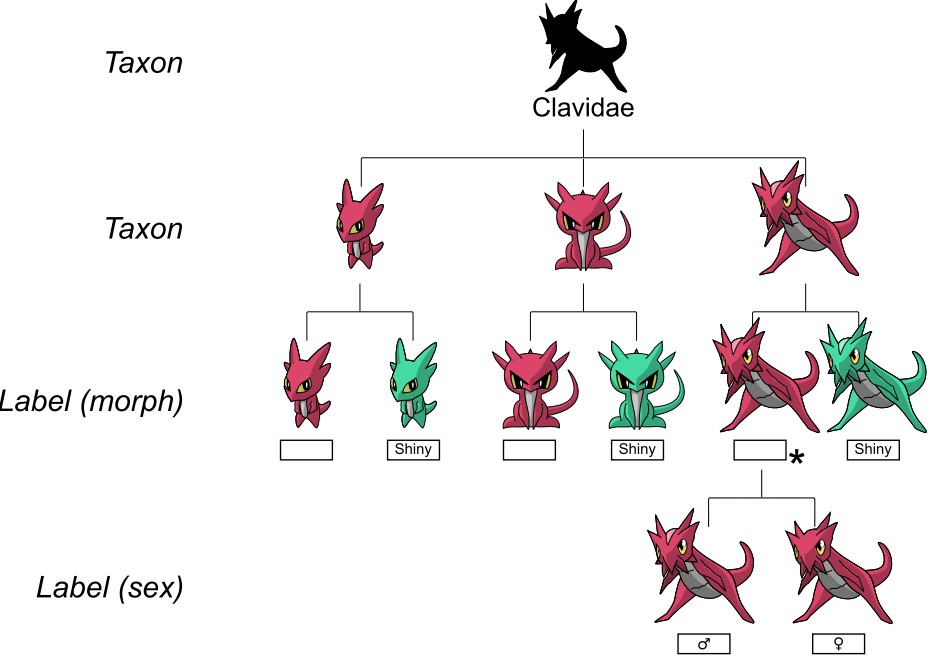

**Fig 3. A taxon tree of the Clavidae: Taxa tagged as endpoints are indicated with an asterisk (***).** This key continues until the correct morph ("Shiny" or default) is known. As the default morph of Clavissima, the species to the right, is defined as an endpoint, no questions regarding the sexes of Clavissima will be asked once its correct morph has been determined.

rare, wingless form, one could make a label for the rarer form (e.g. "wingless form") but use just an empty string for the common form.

The goal of the key is to provide a way to eliminate all but one taxon; the result of the identification process. The key should stop asking questions once the user has narrowed down the possible outcomes to a single endpoint. In a flat list of taxa, every taxon in the list will be an endpoint. In a hierarchical list, the lowermost taxa in the hierarchy will be the endpoints by default. This can be overridden, however, by explicitly tagging a taxon higher up the tree as an endpoint. In this case, information on lower taxa may be displayed if identified while using the key, but no additional questions are asked once the endpoint has been determined. Units below the endpoint level can be used, for instance, to display an image of the relevant subgroup for a taxon, so that the taxon images reflect the input from the user.

*Example*: *Take a taxon tree that consists of three species within a single family. These species can each have a normal form and a rare morph, and the key creator decides that the goal of the key is to find the correct morph. A morph can still have further subdivisions into, say, sexes, but because the parent morph is tagged as an endpoint, the user will not be asked further questions to determine the sex once the morph has been determined. See* Fig 3 *for a graphical representation of this using Clavidae, and lines 25–94 in the supporting information (*S2 File*) for a JSON example.*

## Characters, statements and frequencies

The core element of any Clavis key is the statement. It defines a property of a taxon, separating it from other taxa that have conflicting statements. In JSON code, a single statement takes the following form, in this case stating that birds always have feathers.

```
{
  "id": "statement:birds_have_feathers",
  "taxon": "taxon:aves",
  "character": "character:has_feathers_or_not",
  "value": "state:has_feathers",
  "frequency": 1
}
```

One of the possible values for a statement (the relationship between a taxon and a character) is the id of a state. A statement also may have a frequency: the proportion of cases where individuals of this taxon have this state for this character. It can be set to 0 or 1 or any value in between.

In the default setting, the states of a character are interpreted as being mutually exclusive. If there is a character with states "blue" and "red", a specimen may be either blue or red, but not both at the same time. This means that a taxon that is noted as always being blue is known never to be red, and vice versa. This default behavior can be overridden, however, by defining the character as non-exclusive. In that case, stating that a specimen is blue does not exclude the possibility that it is also (partially) red.

*Example*: *A default statement linking a species to a character state with frequency 1 denotes a species that always has that state. See lines 261–267 in the supporting information (S2 File) for a JSON example.*

*Example*: *A default statement linking a species to a character state with frequency 0 denotes a species that never has that state. See lines 268–274 in the supporting information (S2 File) for a JSON example.*

*Example*: *A default statement linking a species to a character state with frequency 0.5 denotes a species that has that state in 50% of the cases. See lines 275–281 in the supporting information (S2 File) for a JSON example.*

*Example*: *A non-exclusive character can be used to link several states, for instance for several colors, to a taxon. This denotes that that species has all of these colors simultaneously. See Fig 4 for a graphical representation using Clavidae, and lines 177–203 in the supporting information (S2 File) for a JSON example.*

## Multilingualism

Rather than specifying a single language of the key, one can make a key multilingual by specifying an array of any number of ISO 639–1 codes of languages included in the key. When doing so, strings within the key that differ between languages have to be given as localizedStrings; objects containing a version of each of the supported languages. To support different script types, also strings like people's names support localizedStrings.

Not only strings can have different versions for different languages. Links to external online resources can refer to separate language versions (localizedUrls), and images (e.g. containing text) can have different versions for different languages too (localizedMediaElements).

*Example*: *A creator name requiring different transcriptions within a multilingual English/ Ukrainian key*

```
"language": ["en", "uk"],
"creator": "person:wouterkoch",
"persons": [
  {
    "id": "person:wouterkoch",
    "name": {
```

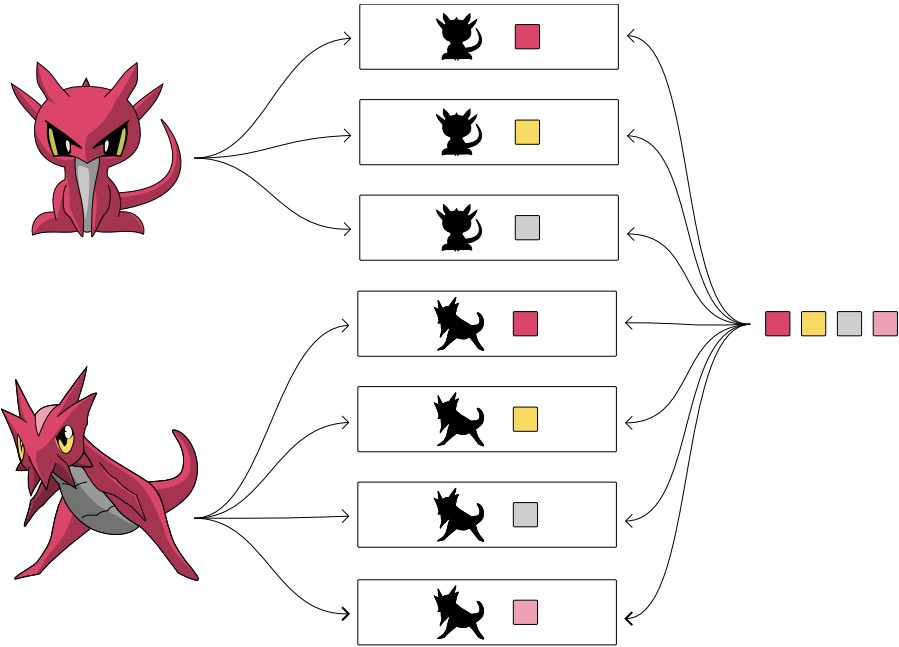

**Fig 4. The colors of Clavis and Clavissima: The species Clavis and Clavissima both have several colors.** In its default setting, the color character will allow the user to select only one color, which will imply that the individual does not have the other colors. By defining the character as non-exclusive, the user can pick and/or exclude freely from the list of possible colors.

```
        "en": "Wouter Koch",
        "uk": "Ваутер Кох"
      }
    }
  ]
```

*Example*: *A creator name that needs no translation, and a character that does, within a multilingual English/Dutch key*

```
"language": ["en", "nl"],
"creator": "person:wouterkoch",
"persons": [
  {
    "id": "person:wouterkoch",
    "name": "Wouter Koch"
  }
],
"characters": [
  {
    "id": "character:eye_color",
    "title": {
      "en": "Color of the eyes",
      "nl": "Kleur van de ogen"
    },
    "states": [
      {
```

```
            "id": "state:blue_eyes",
            "title": {"en": "Blue", "nl": "Blauw"}
          },
          {
            "id": "state:red_eyes",
            "title": {"en": "Red", "nl": "Rood"}
          }
        ]
      }
    ]
```

*Example*: *An organization with names and urls within a multilingual Norwegian/English key*

```
"language": ["no", "en"],
"publisher": "organization:ntnu",
"organizations": [
  {
    "id": "organization:ntnu",
    "name": {
      "no": "Norges teknisk-naturvitenskapelige universitet",
      "en": "Norwegian University of Science and Technology"
    },
    "url": {
      "no": "https://www.ntnu.no",
      "en": "https://www.ntnu.edu"
    }
  }
]
```

## Persons and organizations

Persons and organizations can have multiple roles in different contexts. A person can be a contact person for a key, and one or more persons can be creators or contributors to a key. Persons can also be creators or contributors to media files. A person can have one or more organizations as their affiliation, and organizations can have a person as a primary contact. An organization can be the publisher of a key or media file, and the primary contact for a key can also be an organization instead of a person. Persons and institutions can have resources such as urls and mediaFiles connected to them, e.g. institutional websites, portraits and logos. By defining such entities separately, there are no duplicates to maintain, making the file more compact.

*Example*: *Definition of a person and an organization. See lines 11–23 in the supporting information (S2 File) for a JSON example.*

## Geographic and taxonomic scope

Geographic regions can be defined using a name and/or GeoJSON MultiPolygon. On the top level, a "geography" field can be specified to indicate what geographic region the key covers. One can also define a geography at the taxon or statement level, to provide information on where the taxon occurs, and in which region a taxon has that particular relationship to a character. Within a region specified as the geography of a statement, the statement with a geography takes precedence over the conflicting statement without a stated geography.

*Example*: *A species only occurring in a certain region can have this region specified so this can be taken into account, for example combined with the user's current location. See lines 95–146 in the supporting information* (S2 File) *for a JSON example.*

*Example*: *A certain property of a species may occur at a higher rate in a particular area, which can be denoted using a statement with this higher frequency and a location. This statement will then override the default statement (with the lower frequency) in that area. See lines 299–344 in the supporting information* (S2 File) *for a JSON example.*

## Logical premises

Not all characters are meaningful in every context, and it is sometimes desirable to be able to specify conditions that must be met before a character is shown to the user. Such a condition may be that the user has given a certain answer to some other character in the key first. To this end, characters can have a logical premise, specifying which facts have to be established for it to be relevant. Logical premises can also relate several facts combined, including numerical values, and consist of strings using unary and binary operators in JavaScript notation.

Logical premises are seldomly required to make a key work. If characters are scored only for those taxa that they are relevant for distinguishing, this in itself will normally ensure that the characters stay hidden from the user until they are relevant to show. However, a situation may arise where some of the taxa in the key are polymorphic, i.e. if they either may or may not have a certain property. In this situation, a character referring to this property will normally become visible to the user if all the remaining taxa *can* have it, which will likely cause confusion if the actual specimen being identified does *not* have it. By introducing a logical premise, such a character can be hidden until the user has explicitly indicated that the specimen does in fact have the property.

*Example*: *Species may be determined by the property of their wings, but some of these species may also have a wingless form. Characters pertaining to wings should only be visible once it has been established that wings are present. Other species may be determined by the properties of their nests, but one can encounter a specimen not at its nest. The availability of the nest has to be determined before asking about nest characteristics. See lines 220–248 in the supporting information* (S2 File) *for a JSON example.*

## Numerical values

The possible values that a character can have may take the form of a set of discrete states, such as "present"/"absent", or "red"/"blue". They may, however, also come in the form of numerical values, such as counts or measurements. In a key this can be implemented with a character of the type "numerical". Numerical characters specify a minimum and maximum value that the measurement can have, a step size and its unit. Statements specifying values for numerical characters can define a range, or a single value. If more advanced metrics are needed for the key, such as a probability function over the numerical values for use in Bayesian analysis, this can be implemented through a call to an external service (see below).

*Example*: *The weight of a species may vary, both between individuals within a species and between species. To distinguish species based on their weight, it can be added as a numerical character, where weights can be specified for example in whole kilograms within a given range. A statement can then be included to register the possible weight range of a given taxon. See lines 249–258 (character) and lines 289–298 (statement) in the supporting information* (S2 File) *for JSON examples.*

## External services

Various units in a key may be subject to changes that are ideally managed outside the key, in designated centralized systems. Taxon names can change, as can urls to supplemental information, media files, geographic ranges etc. It is best practice to not duplicate such resources, but to harvest these via an Application Programming Interface (API) or other service interface. To facilitate this, Clavis allows the specification of external services, where documentation for use of the service can be linked. Various units in the key, such as taxa, can refer to such external services through one or more externalResources, connecting the service and the relevant external id to the taxon at hand.

External services should not contain information critical to the workings of the key, as not every implementation can be expected to contain the necessary code to retrieve the information the service provides. External services are useful, however, for features steering presentation, and can provide complex information that depends on various user inputs and other contextual information. To sort taxa by probability, for example, an external service may provide probability scores for taxa based on the geographic location of the user, the season, and properties like coloration and size of the specimen. An interface that does not implement calling this service will simply not sort the taxa by probability, but will still be fully functional.

*Example*: *To allow for features such as the retrieval of the name of a taxon, one can use calls to a taxonomic nomenclature service. See lines 432–436 (service) and lines 47–50 (externalResource) in the supporting information* (S2 File) *for JSON examples.*

*Example*: *A hypothetical service may return the probability for a provided taxon to occur, given the location and size of the specimen. See lines 437–443 in the supporting information* (S2 File) *for a JSON example.*

## Required expertise

Some characters are harder to evaluate than others and may even require special equipment. To warn and assist users, a key can contain userRequirements, describing such required skills or tools. These requirements can be connected to characters so that the user might filter out characters requiring skills or equipment they do not have, be presented with additional info to complete the task, or simply be warned.

*Example*: *Taxonomists may use various tools and methods, e.g. handling the specimen in a certain way, dissection, microscopy, that require skills beyond simple visual inspection. The user can be made aware of these required skills, so that they can obtain them (for instance through a description containing a guide connected to this userRequirement), or opt to skip the question as they cannot evaluate it, for example when identifying from a picture. See lines 249–258 (character) and lines 417–429 (userRequirement) in the supporting information* (S2 File) *for JSON examples.*

## Media elements

Illustrations greatly improve the usability and aesthetics of any key. Most entities in a Clavis key can contain a reference to one or more images or other media elements. Media can be linked to taxa, characters and states, but also to persons, organizations, userRequirements, statements, etc. To achieve this, Clavis defines mediaElements, containing one or more mediaFiles. These mediaFiles can contain and/or link to image, sound, or video files, using an array of mediaFiles to allow different sizes of the same file to be included. As states do not require a title, more visual keys can contain states with only an image.

*Example*: *States in a character relevant to a number of species can each have a drawing or other pictures attached to them so that the user can better evaluate the character. See lines 207–*

*218 (states) and lines 446–496 (mediaElements) in the supporting information* (S2 File) *for JSON examples.*

## Descriptions

There are many aspects of an identification key where a more elaborate description is desirable or necessary. To this end, many elements can have a short and/or extended description in valid markdown notation, as well as a url to an online description.

**Example**: *A description can be used to explain to the user what they need to do to evaluate a certain character, such as counting the teeth of a mammal. There can also be an additional link to an external article on tooth formulas. See lines 422–429 in the supporting information* (S2 File) *for a JSON example.*

## Followup keys

Once a key has identified a taxon as far as it is intended to, i.e. when it has narrowed it down to an endpoint taxon, the user is presented with the result. It may however be possible to further determine the result. If a key exists somewhere that can help with this, it can be referred to as a followup key from the relevant taxon. It can be either a url, or a reference to an (external) service. This feature can also be used to split keys into several smaller keys that refer to one another through this mechanism.

**Example**: *A key identifying to the family level can advise the user that a different key is available to determine the species within that family, and link to it. See line 56 in the supporting information* (S2 File) *for a JSON example.*

## Conclusions

The examples provided here illustrate the versatility of Clavis as a key format. Several of its features are, to our knowledge, not supported by any other format, nor is the totality of its features.

A crucial aspect in identifying any taxon is the geographic origin of the target specimen. Primarily, it dictates which taxa are candidates for its identity, and thus which key(s) can be used for it and which taxa within the keys are to be considered. Secondarily, it dictates the possible traits of the taxa, insofar as these vary geographically. By referring to external services for geographical information, keys can be made to directly benefit from Species Distribution Models hosted elsewhere, ever improving as more data and improved methodology become available.

It is important to realize that keys are designed to function as a whole, and that its contents need to be regarded in context of the key. Characteristics defined and scored in the context of distinguishing between taxa within a limited taxonomic and geographical scope can be misleading when put in contexts other than that of the key. It may in some cases be possible to extract traits from a key for use in a different context, such as for a taxon diagnosis or a trait database. Taxonomic knowledge is required to assess the relevance of such traits outside of the key's context, however.

A particularly potent application of the Clavis format will be an implementation in tandem with automated image recognition. Since statements are stored as separate entities, keys are polythetic [17]; there is no fixed path through the key requiring that taxa are evaluated against their characteristics in any particular order. This means that the key can be applied to any subset of the taxa as easily as to the full taxon set. Which characters are displayed to the user will automatically adjust according to the subset of taxa. This allows for a reduction of the probable identifications by a machine learning algorithm as a first step, followed by keying of the

relevant subset of taxa to make a final determination. This mechanism potentially reduces the need for much of the user input of a full key, thus saving the user considerable effort and reducing the possibility of the user providing incorrect input that is inherent each time user input is provided. Conversely, it reduces the reliability solely on machine learning for identification, providing a mechanism of quality control of the algorithm output, and an opportunity for the end user to learn a great deal more than they would with only a recognition model prediction. Alternative polythetic key formats can be used in a similar way, but do not offer the entire feature set, openness and native readability that Clavis does (see Table 1).

The development of Clavis has been done in close collaboration with taxonomic experts. While this has enabled us to include many diverse features covering needs that have arisen in the past, no such endeavor can expect to produce a final version guaranteed to cover all needs. Further adoption may also bring to light ambiguities or shortcomings that will need to be addressed. Our aim is to continue to update the format, releasing new iterations with improvements. Use of previous versions will remain possible, and we aim to maintain backwards compatibility wherever possible. We invite the community to contribute to the development of Clavis, through submitting issues on GitHub, and resolving issues by answering questions or proposing code changes through pull requests. We hope that solutions supporting Clavis, be it key building software or end-user interfaces, both of which we plan to create examples of (see Fig 5), will be shared openly as part of a broader ecosystem of use and re-use.

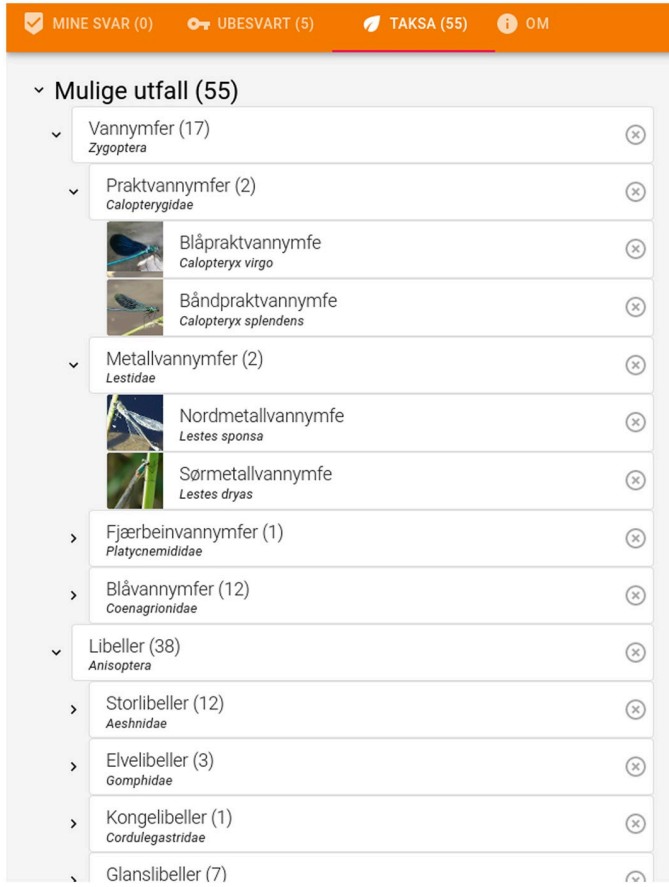
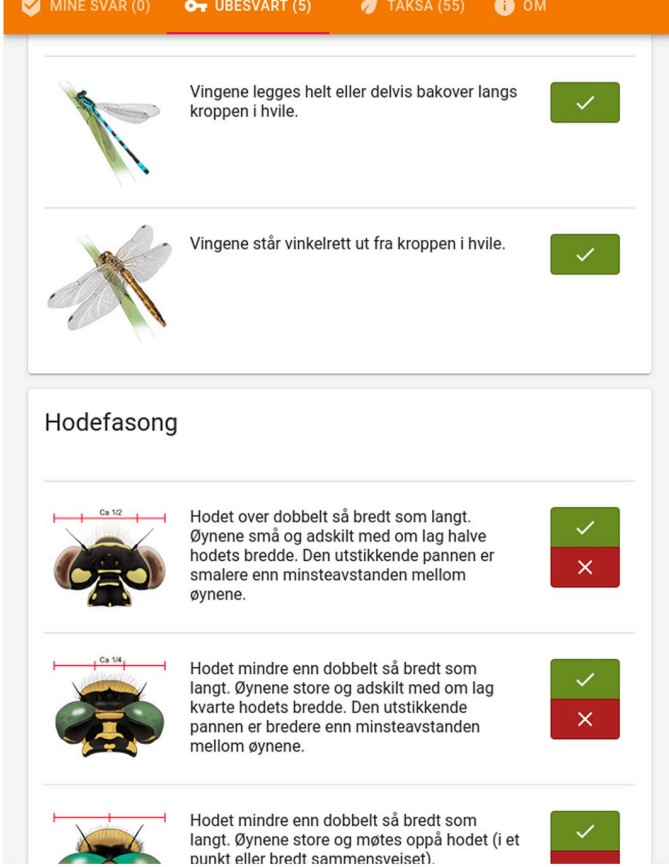

**Fig 5. An example implementation of a Clavis key graphical user interface.** Based on a working Odonata key previously published by the Norwegian Biodiversity Information Centre. A hierarchical list of taxa (left) is reduced by providing input on characters (right). Only characters relevant for all remaining taxa are shown. Drawings CC BY 4.0 Hallvard Elven.

Our aim is for Clavis to be a manner of storing the taxonomic knowledge needed for identification in a way that allows for the representation of the complexity and nuances inherent to such knowledge. The open exchange of taxonomic knowledge, unambiguously captured with as much of the auxiliary details needed for its application, is essential for the preservation of invaluable, increasingly elusive knowledge. Through this extensive formal definition of how to unambiguously capture such knowledge, we hope to provide the groundwork for software tools that serve the needs of both creators and users of keys of various complexities.

We believe that the storage of taxonomic knowledge with the level of detail exemplified here, combined with user interfaces making it accessible, is vital in enabling observers to gather the data needed for apt nature management. Particularly within citizen science, the potential of tools built on an identification key format such as Clavis is considerable. The more accessible this expert knowledge is, the more accurate the identifications made by the user will be, and keys on generally less well-known taxa can aid in closing the taxonomic gaps in the data corpus. The possibility of storing the user input together with a reported observation can provide important metadata on the identification and its quality. The results of these projected advances in data collection quality feed back into the areas where these data are used, from research and spatial distribution models to the decision making processes related to the biodiversity crisis in a changing world.

## Supporting information

**S1 File. Clavis JSON-schema.** The formal definition of what constitutes a Clavis-compliant key.
(ZIP)

**S2 File. Clavis key example: Pokémon.** A Clavis-compliant key to a number of Pokémon. Serves to illustrate all the different aspects that Clavis supports, rather than to provide a fully functional and complete key.
(ZIP)

**S3 File. Clavis key example: Titmice.** A Clavis-compliant key to Norway's titmice (Paridae). Serves as a real life example of a fully functional and complete key, using only a selection of Clavis' capabilities.
(ZIP)

## Acknowledgments

We are grateful to Askild Aaberg Hofsøy Olsen for his feedback on the technical aspects of the Clavis schema.

## Author Contributions

**Conceptualization:** Wouter Koch, Hallvard Elven, Anders G. Finstad.

**Methodology:** Wouter Koch.

**Supervision:** Anders G. Finstad.

**Visualization:** Wouter Koch.

**Writing – original draft:** Wouter Koch, Hallvard Elven, Anders G. Finstad.

**Writing – review & editing:** Wouter Koch, Hallvard Elven, Anders G. Finstad.

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
