## [Decision Letter · Decision Letter 0]

7 Sep 2022

PONE-D-22-15296Clavis: an open and versatile identification key formatPLOS ONE

Dear Dr. Koch,

Thank you for submitting your manuscript to PLOS ONE. After careful consideration, we feel that it has merit but does not fully meet PLOS ONE’s publication criteria as it currently stands. Therefore, we invite you to submit a revised version of the manuscript that addresses the points raised during the review process.

Three external reviewers have now evaluated your submission. They have identified a number of concerns that need to be carefully addressed when preparing your revisions. Please respond to all of the points they have raised, paying particular attention to the reviewers' requests for methodological clarifications and their suggestions for opportunities to improve the utility and accessibility of the tool.

We look forward to receiving your revised manuscript.

Kind regards,

Jamie Males

Editorial Office

PLOS ONE

Journal Requirements:

2. We note that Figure 3, 4, and 5 in your submission contain copyrighted images. All PLOS content is published under the Creative Commons Attribution License (CC BY 4.0), which means that the manuscript, images, and Supporting Information files will be freely available online, and any third party is permitted to access, download, copy, distribute, and use these materials in any way, even commercially, with proper attribution. For more information, see our copyright guidelines: http://journals.plos.org/plosone/s/licenses-and-copyright.

1. You may seek permission from the original copyright holder of Figure 3, 4, and 5 to publish the content specifically under the CC BY 4.0 license. 

3. We note that you have stated that you will provide repository information for your data at acceptance. Should your manuscript be accepted for publication, we will hold it until you provide the relevant accession numbers or DOIs necessary to access your data. If you wish to make changes to your Data Availability statement, please describe these changes in your cover letter and we will update your Data Availability statement to reflect the information you provide

Reviewers' comments:

Reviewer's Responses to Questions

**Comments to the Author**

1. Is the manuscript technically sound, and do the data support the conclusions?

Reviewer #1: Partly

Reviewer #2: Yes

Reviewer #3: Yes

2. Has the statistical analysis been performed appropriately and rigorously? 

Reviewer #1: N/A

Reviewer #2: N/A

Reviewer #3: N/A

3. Have the authors made all data underlying the findings in their manuscript fully available?

Reviewer #1: Yes

Reviewer #2: Yes

Reviewer #3: Yes

4. Is the manuscript presented in an intelligible fashion and written in standard English?

Reviewer #1: Yes

Reviewer #2: Yes

Reviewer #3: Yes

5. Review Comments to the Author

Reviewer #1: The manuscript presents a format to represent the information needed to build an interactive identification key based on JSON format.

Although novel proposals are made in the manuscript, such as the use of JSON and its capabilities to represent geographic regions through polygons, and to specify characters and states in multiple languages, it lacks a solid methodology in the construction of the proposal, as well as of theoretical foundations of taxonomy and biological identification.

Introduction

It is stated that “A number of digital identification key formats exist [15, 16], but these come with a number of limitations in what they can represent”, but no explanation of these limitations is given.

Methods

The method is poor, as it does not delve into taxonomic concepts nor does it explicitly state how the Clavis format was defined. Character types and character states and the problems of their computer representation are not listed or discussed in depth.

A comparative scheme is needed to show the advantages and strengths of the “Clavis” representation against other popular formats such as the Delta format.

Results

Using a fictitious taxonomic group (Pokémon in this case) can provide didactic benefits in certain contexts, but in the case of showing the power of representation of a representation structure of taxonomic knowledge it may not be adequate, since it does not give the opportunity to address the range of peculiarities and exceptions observed in real taxa.

Discussion

It is stated that “A particularly potent application of the Clavis format will be an implementation in tandem with automated image recognition” because “there is no fixed path through the key requiring that taxa are evaluated against their characteristics in any particular order. This means that the key can be applied to any subset of the taxa as easily as to the full taxon set”, two important omissions are made: a) it is not referring to the relevant theoretical framework, namely the polythetic identification, and b) that other formats, such as Delta, can be used in the same way.

Reviewer #2: Dear Editor & Authors

This manuscript (PONE-D-22-15296), “Clavis: an open and versatile identification key format” provides a description of a new tool similar in purpose (though not in function) to matrix-based multiple-access keys for identifying unknown organisms. The paper walks the reader through the rationale for digital keys broadly, and then discusses the basic design elements of the Clavis key, which is implemented in the JSON language.

In general, the manuscript is clearly written, and the figures (primarily schematics illustrating workflows) are clear.

The authors provide, as appendices and via a GitHub link, samples of keys that could be viewed as test cases and, presumably, templates that a reader could use to build their own keys. I was unable to figure out how to execute the code for the sample keys and so didn’t review those.

In general, I strongly support the publication of this new tool. I have some suggestions for improvement, but they are all with the text, not with the quality or substance of the tool itself. I think that the more tools that are out there, the better- especially things like this that are freely available and published in an open access journal. It’s not entirely clear to me how substantially this contribution will move the field forward (see comments below), but I might not be the target audience and I think tool that the authors have put together definitely should be put out into the world so that it can find its audience.

I have a couple general comments/suggestions for improvement below. These are followed a few more specific comments with line numbers.

General comments

1) It wasn’t clear to me who the audience was. This normally is not something I think about in a review, but I found myself struggling with it here. There seems to me to be two potential audiences: 1) people with the coding expertise to use this work as a template to develop similar tools for their own use and 2) taxonomists who are interested in developing digital keys and have at least some background in coding but who might need to collaborate with someone with more coding expertise to make this work(I fall into the latter category). It may be that the paper would be quite useful to the first audience as written, but for the second audience (if taxonomists were one of the intended audiences), it needs some improvement to convincingly make the case that it’s worthwhile for a taxonomist ot invest the time in learning how to use the tool.

Specifically, here are the issues that could use further elaboration, clarification, etc.

2. There needs to be a quick guide or something (just a one or two pager, or maybe something like an R vignette). After some internet searching, I’m embarrassed to say that I couldn’t even figure out how to execute the .json files. That’s partly on me as the reader for not being savvy enough, but I have an R console open daily and am relatively comfortable with coding & GitHub etc. If I can’t figure it out, I think it’s likely that many taxonomist types also will have trouble. So a quick guide to just executing the file in the download folder would be an easy and substantial improvement.

3. From the taxonomist side, it wasn’t entirely clear what the improvements of this approach are over more traditional matrix-based keys (see specific comment below). I could see that you were making a case, but it would benefit from thinking about the taxonomist audience (who like me, think all the time that it would be nice to build an interactive key but don’t have the time and/or expertise so just write a standard dichotomous key) and more clearly outlining the specific benefits of this approach.

4. In the ‘Format overview’ section- I didn’t see what the benefit (if any) of some of the features would be to the taxonomist (e.g. Person, External Service)

5. In the introduction (line 84), it was briefly mentioned that there ought to be an interface so that a taxonomist could just put in the content without manually writing the code, but it appears that this interface doesn’t yet exist. If this tool is going to be of value to most taxonomists, it would seem that this kind of interface would be essential.

Below are specific line edits. Some of these are redundant with or are special cases of the general comments listed above.

Line 84- “This article …”

I find this problematic- why announce a new tool but not produce the new tool for use?

Line 87-“… are illustrated here using fictional taxa...”

I think most taxonomists would agree that there are two issues with traditional paper keys.

1) the difference between dichotomous and ‘multiple access’ keys, which the authors are clearly addressing here with Clavis.

2) there is extensive variation within species (I’m revealing my training as a plant systematist here). I don’t know anything about Pokemon, but I’m assuming there isn’t the analog of variation within taxa among Pokemon that would impede confident identification.

I see the pedagogical value in dealing with taxa that have no within-taxon variation and no complicated taxon-specific jargon, but useful new identification tools can’t pretend those challenges (issue #2) don’t exist.

Clavis is obviously meant to deal with issue #1 and not #2. However, an honest conversation about the building more accessible identification tools can’t ignore #2. Forward progress in this area needs to at least acknowledge #2 (because it’s one of the main reasons traditional keys don’t work) even if the progress toward #1 isn’t addressing #2 specifically.

Line 158- For this paragraph (or possibly elsewhere if you think it would fit better somewhere else), the authors should make a stronger and/or more clear argument about the advantage of a JSON based key over a more traditional matrix-based key. I feel like I may be missing a key point here that the authors perhaps are not making clearly enough. Wouldn’t it be possible to build a traditional matrix-based key around a sparse dataset by just having a lot of NAs? I feel as though the argument is that this Clavis approach is somehow different from that, but as written, the message of the important innovation provided by Clavis isn’t immediately apparent. Perhaps some clarifying summary language about the way JSON can articulate the different pieces of information in a way that a traditional matrix-based key can’t would help strengthen the authors’ case.

Line 166- “…generally provide taxonomists with a key editing interface…”it sounds like what you’re saying here is that a key editing interface is necessary for a taxonomist to be able to build a key, but it doesn’t appear that you’re providing this key editing interface. If that’s incorrect, you could clarify. If that’s correct, then it’s not clear what the point is of producing only half of the required tools

Line 578- “…not supported by any other format…”- the authors need to spell out exactly what these differences are. Perhaps they are clear to someone proficient in JSON. As a taxonomist who has used a modest number of different styles of digital keys (but never built one myself), it’s not clear how this one is different and what the argument is that it’s superior to alternatives (I’m not saying that it isn’t good- I’m saying that it’s your responsibility to more clearly state how/why it’s good)

Reviewer #3: This paper presents Clavis, a schema (written in JavaScript Object Notation) for storing and sharing biological species information. The schema has great potential due to its open format and versatility. If received by the scientific community as intended by the authors, it can become a go to format for storing and disseminating various biological species data.

The paper is very well written, and concepts are accompanied by useful visual and textual examples. Three example schemas are also included as supplementary material.

Some general comments:

The Clavis schema is presented in this paper as identification key software when the information given is focussed primarily on the data storage and sharing function of the schema. This disconnect is problematic and may be solved by including examples of both a key building and end user interface (figure 5 shows a basic example). It will go a long way in showcasing the practical application of the schema as well as to encourage use by other researchers. As the authors include three schema examples in the supplementary materials, they can presumably present the key building software that they used.

Although the example schemas seem relatively easy to compile, they are simple, while real world schemas may be much more complex and compiling them may be a challenge for some users. As suggested by the authors, interactions with the Clavis community on platforms such as Github will be very helpful for trouble shooting in this respect.

The use of images in identification resources are important and will become more so in the future. Although the authors state that it is possible to link media files to taxa and characters, it is not clear to what degree images can be included directly in the end-user interface.

An interpretation from the text is that the schema can do translations. Is this correct or does it only allow one to specify which language is being used for a specific input? It might be good to clarify this in the text. In line 376 the authors state that “one can make a key multilingual by specifying an array of language codes that are supported”. How many languages are supported?

Can taxon names be linked to external online databases so that they update automatically in the key/schema when changed on said database? This will be a very useful feature allowing the data to stay up to date without manual editing every time a taxon name changes (which happens relatively often in some species groups).

Overall, the paper is a valuable contribution to the journal and to science and I suggest it be accepted with minor revisions.

6. PLOS authors have the option to publish the peer review history of their article (what does this mean?). If published, this will include your full peer review and any attached files.

Reviewer #1: **Yes: **Miguel Murguía-Romero

Reviewer #2: No

Reviewer #3: No

---

## [Author Response · Author response to Decision Letter 0]

17 Oct 2022

We thank the reviewers for their useful comments, and the substantial improvement this has brought to the manuscript. We have addressed each of the points raised by the reviewers and the editorial office. For your convenience we add a latexdiff pdf where all edits are marked.

Below we give a general point by point reply to the recurrent themes in the comments of the reviewers and the editor, grouped by general issue raised:

The lack of an explicit connection to taxonomic theory made it unclear for taxonomists if the proposed format covers taxonomic complexity and thus how relevant it is. This was an oversight from our part (all with a primary background in biology and/or taxonomy) which we have addressed throughout the manuscript. It is vital to the readers' understanding that the messy reality of real world taxonomy and how we incorporated it in Clavis is acknowledged early in the manuscript, which it now is.

The distinction between the data format and an implementation using the format remained unclear, and where it was clear the rationale behind it was not stated clearly enough, leaving the impression our proposal is "incomplete". We address this view explicitly, laying out why we consider it important not to conflate the two. We also substituted many instances of technical terms like "data format" and "implement" with terms suitable for a broader readership.

References to Pokémon, chosen to disconnect the examples from any real taxonomy that may lead to confusion if characteristics are or later become subject to debate, had the opposite effect. Lack of familiarity with these fictional creatures, while never required, made readers feel they cannot fully understand the examples. We understand and share this view, and have made the examples in the text generic without referencing any Pokémon. They are now solely used in the appendix, for code exemplification alone.

---

## [Decision Letter · Decision Letter 1]

3 Nov 2022

Clavis: an open and versatile identification key format

PONE-D-22-15296R1

Dear Dr. Koch,

We’re pleased to inform you that your manuscript has been judged scientifically suitable for publication and will be formally accepted for publication once it meets all outstanding technical requirements.

Kind regards,

Hong Qin

Academic Editor

PLOS ONE

Additional Editor Comments (optional):

Reviewers' comments:

Reviewer's Responses to Questions

**Comments to the Author**

1. If the authors have adequately addressed your comments raised in a previous round of review and you feel that this manuscript is now acceptable for publication, you may indicate that here to bypass the “Comments to the Author” section, enter your conflict of interest statement in the “Confidential to Editor” section, and submit your "Accept" recommendation.

Reviewer #1: All comments have been addressed

Reviewer #3: (No Response)

2. Is the manuscript technically sound, and do the data support the conclusions?

Reviewer #1: Yes

Reviewer #3: Yes

3. Has the statistical analysis been performed appropriately and rigorously? 

Reviewer #1: N/A

Reviewer #3: N/A

4. Have the authors made all data underlying the findings in their manuscript fully available?

Reviewer #1: Yes

Reviewer #3: Yes

5. Is the manuscript presented in an intelligible fashion and written in standard English?

Reviewer #1: Yes

Reviewer #3: Yes

6. Review Comments to the Author

Reviewer #1: The authors have modified the main example by focusing on a real taxonomic group (and not a fictitious taxonomic group; Pokémon in this case) showing the representational power of their proposal.

The authors reinforce the theoretical framework, including the concept of polythetic identification.

Now a comparison table showing the advantages and strengths of the “Clavis” representation against other popular formats such as the Delta format is included.

Reviewer #3: The authors have adequately addressed all of the previous comments and suggestions. One aspect that stands out is the lack of user interface examples. The authors have clarified that the decision not to include these is conscious as to not limit the scrutiny of the schema itself. In this case I feel that the title (Clavis: an open and versatile identification key format) may mislead the reader, who might expect to find user interface examples of an identification key. Perhaps a slightly modified title such as “Clavis: an open and versatile data format for identification keys”, that emphasizes Clavis as a data format and not an actual identification key, should be considered.

While not one of my original critiques, the current fictional taxa used in the examples are perhaps more appropriate than the Pokémon characters used in the previous manuscript.

All in all, this work is valuable and novel and I look forward to seeing its implementation in the future.

7. PLOS authors have the option to publish the peer review history of their article (what does this mean?). If published, this will include your full peer review and any attached files.

Reviewer #1: **Yes: **Miguel Murguía Romero

Reviewer #3: No

---

## [Editor Report · Acceptance letter]

8 Nov 2022

PONE-D-22-15296R1 

Clavis: an open and versatile identification key format 

Dear Dr. Koch:

I'm pleased to inform you that your manuscript has been deemed suitable for publication in PLOS ONE. Congratulations! Your manuscript is now with our production department. 

Kind regards, 

on behalf of

Dr. Hong Qin 

Academic Editor

PLOS ONE